# Chiral Recognition of Hexahelicene on a Surface via the Forming of Asymmetric Heterochiral Trimers

**DOI:** 10.3390/ijms20082018

**Published:** 2019-04-24

**Authors:** Hong Zhang, Hong Liu, Chengshuo Shen, Fuwei Gan, Xuelei Su, Huibin Qiu, Bo Yang, Ping Yu

**Affiliations:** 1School of Physical Science and Technology, ShanghaiTech University, 393 Middle Huaxia Road, Pudong, Shanghai 201210, China; zhanghong1@shanghaitech.edu.cn (H.Z.); liuhong@shanghaitech.edu.cn (H.L.); ganfw@shanghaitech.edu.cn (F.G.); suxl@shanghaitech.edu.cn (X.S.); 2Shanghai Institute of Ceramics, Chinese Academy of Sciences, Shanghai 200050, China; 3University of Chinese Academy of Sciences, Beijing 100049, China; 4School of Chemistry and Chemical Engineering, State Key Lab of Metal Matrix Composites, Shanghai Jiao Tong University, Shanghai 200240, China; shenchengshuo@sjtu.edu.cn

**Keywords:** chiral recognition, helicene, assembly, STM, nc-AFM

## Abstract

Chiral recognition among helical molecules is of essential importance in many chemical and biochemical processes. The complexity necessitates investigating manageable model systems for unveiling the fundamental principles of chiral recognition at the molecular level. Here, we reported chiral recognition in the self-assembly of enantiopure and racemic hexahelicene on a Au(111) surface. Combing scanning tunneling microscopy (STM) and atomic force microscopy (AFM) measurements, the asymmetric heterochiral trimers were observed as a new type of building block in racemic helicene self-assembly on Au(111). The intermolecular recognition of the heterochiral trimer was investigated upon manual separation so that the absolute configuration of each helicene molecule was unambiguously determined one by one, thus confirming that the trimer was “2+1” in handedness. These heterochiral trimers showed strong stability upon different coverages, which was also supported by theoretical calculations. Our results provide valuable insights for understanding the intermolecular recognition of helical molecules.

## 1. Introduction

Chirality [1], one of the fundamental properties of molecules with handedness, has played a vital role in physics, chemistry, and biology. In particular, how chiral molecules cooperate and recognize with each other is an essential issue in both asymmetric synthesis and various biochemical processes. In order to simplify the complexity of the problem, a prototypical model system such as helicene [2], which is a promising candidate for future optoelectric and spintronic devices [3,4], has been intensively investigated to unveil the supramolecular interaction principles among chiral molecules. 

Notably, due to its high spatial resolution [5,6] and single molecule manipulation capabilities [7], scanning tunneling microscopy (STM) has evolved as a valuable technique for understanding the intermolecular recognition of helicenes at surfaces [6]. For example, the heterochiral dimers of heptahelicene ([7]H) on Cu(111) are decomposed into two isolated molecules by STM manipulation, then their handedness is determined by STM imaging [8]. In addition to STM, noncontact atomic force microscopy (nc-AFM) has proven to be another indispensable tool for characterizing the molecular handedness at a submolecular level. One recent representative work explored the [7]H derivative on Ag(111), in which the submolecular structure and handedness of the enantiomeric products were identified by high resolution nc-AFM with a functionalized tip [9]. Aside from this, nc-AFM imaging is believed to be a promising approach to overcome the difficulty in discriminating the handedness of each molecule in the chiral aggregates and self-assembly structures [10].

Recently, chiral recognition in the two dimensional (2D) self-assembly of helicenes on metal surfaces has attracted a great deal of interest. It is also a modern approach for understanding the fundamental steps in the crystallization of racemates [2,6,11,12,13]. For instance, the self-assembly of [7]H and its derivatives have been studied systematically on various metal surfaces, in both enantiopure and racemic forms (*rac*), revealing that a homochiral trimer or tetramer with three- or four-fold symmetry [14,15,16], and heterochiral dimers constituted the building blocks of the corresponding assemblies [15,17,18]. Moreover, spontaneous resolution was observed for cyano-functionalized *rac*-[7]H on Cu(111) [19]. In addition to [7]H, chiral recognition has also been investigated in the 2D crystallization of pentahelicene ([5]H) on Cu(111), demonstrating that homochiral dimers are the building blocks for the 2D self-assembly [20]. Nevertheless, studies on primitive hexahelicene ([6]H) are still absent, with the exception of a few studies that reported on the 5-amio [6]H system [21,22].

Investigating the chiral recognition and assembly features of primitive [6]H without any influence from the functional groups is of great interest. Here, we present the results of the chiral recognition and assembly of enantiopure and racemic [6]H using combined STM and nc-AFM techniques on an Au(111) substrate.

We observed that heterochiral trimers served as building blocks for *rac*-[6]H self-assembly, in strong contrast with previous *rac*-[7]H or *rac*-[5]H results that showed that heterochiral or homochiral dimer recognitions are favored for van der Waals interactions [8,15,17,18]. We found that this asymmetric trimer was actually a “2+1” type, composed of two homochiral molecules and one enantiomer with the opposite handedness, which is a novel building block in helicene self-assembly structures.

## 2. Results and Discussion

### 2.1. Chirality Determination Method

The molecular structures of *P*-[6]H and *M*-[6]H, which are mirror images of each other, are demonstrated in Figure 1a. The denominators *P* and *M* represent right and left handedness [12], which respectively correspond to the clockwise and anticlockwise descending sequence of the molecular apparent height. The explored [6]H molecules were synthetized by following the reported procedure [23]. A CO functionalized tip was used for the nc-AFM measurements to achieve submolecular resolution [24,25]. To prepare the CO functionalized tip, NaCl was firstly evaporated on the Au(111) surface before evaporating the [6]H molecules [26,27], which were used to pick up CO on the tip. The influence of NaCl on the self-assembly of [6]H was ruled out, as demonstrated in the Appendix A. The molecules were evaporated onto the substrate at room temperature in an ultrahigh vacuum chamber and then cooled down to 5 K for the STM/nc-AFM experiments.

We first demonstrated that the handedness of the isolated [6]H monomer could be well-characterized by combining STM and nc-AFM measurements with the CO functionalized tip used for achieving submolecular resolution [24,25]. The experiment was carried out by taking enantiopure *P*-[6]H for instance. Enantiopure *P*-[6]H molecules were obtained by an enantioseparation process and the absolute configuration was further determined using electrotronic circular dichroism (CD) measurements [28]. At a very low coverage, the isolated *P*-[6]H monomer was easily found on the sample. As the constant current STM image in Figure 1b shows, the *P*-[6]H monomer appeared as a round disk with an off-centered protrusion with a diameter of 13 Å, which was similar to the previously reported STM images of [7]H[6] and [5]H[20]. This indicated that [6]H was also adsorbed with terminal phenanthrene parallel to the Au(111) surface, while the rest of benzene rings spiraled up away from the surface [19]. However, the determination of the handedness was rather difficult due to the STM resolution, since it was hard to tell whether the apparent height descended in a clockwise or anticlockwise manner, as indicated in Figure 1b by two curved arrows. To unambiguously identify the molecular handedness, the nc-AFM experiment was carried out (Figure 1c–e). The non-planarity of [6]H causes the nc-AFM images to show only the most protruding benzene ring of the molecule instead of the whole inner bonding structures [29,30]. Nevertheless, by comparing the nc-AFM images at different tip-sample distances, the descending orientation of the [6]H apparent height was determined. When the tip-sample distance was large, only a dark halo was observed in the nc-AFM image, as shown in Figure 1c, due to the attractive interaction between the tip and the sample. By locating the tip closer to the sample, the protruding part of the outmost benzene ring can be detected in Figure 1d, it is the bright-extended shape in the nc-AFM image indicating the descending orientation of the outmost benzene ring, as marked by an arrow in Figure 1d. Once the tip approached further, as displayed in Figure 1e, the repulsive force was too strong to determine the descending orientation of the outmost benzene ring. Generally, the descending orientation of the outmost benzene ring of isolated *P*-[6]H can be resolved from an nc-AFM image measured at the proper tip-sample distance. Based on both the descending orientation of the outmost benzene ring as indicated by the arrow direction in the nc-AFM image in Figure 1d and the possible clockwise or anticlockwise descending orientationas demonstrated in the STM image in Figure 1b, we concluded that at this apparent height only the clockwise descending orientation of the *P*-[6]H was allowed. This was consistent with the *P*-handedness determination provided by the CD spectra measurements, and proved the feasibility of our approach for characterizing the molecule handedness by combining STM and nc-AFM measurements.

### 2.2. Chiral Recognition of Hexahelicene ([6]H)

To investigate the chiral recognition of [6]H, we first compared the *P*-[6]H and *rac*-[6]H self-assemblies on an Au(111) surface at a relatively low coverage. The most favored chiral aggregates of the *P*-[6]H assembly were homochiral tetramers, as shown in Figure 2a. One representative zoomed-in STM image of a *P*-[6]H tetramer is displayed in Figure 2b, and the nc-AFM images of *P*-[6]H tetramer measured at the proper tip-sample distance are displayed in Figure 2c. The results showed that the tetramer appeared as a four-blade propeller, with each blade representing one helicene. This is different to the previously reported *C*_4_ symmetric tetramer of enantiopure [7]H on Cu(100) at all coverages and on Au(111) at high coverage with 0.95 monolayer [10]. The observed *P*-[6]H tetramer shows *C*_2_ symmetry in a rhombus shape with interior angles of about 120° and 60°, which might be influenced by the symmetry of the Au(111) substrate. In contrast to the tetramers of *P*-[6]H, asymmetric trimers were observed for *rac*-[6]H with two mirror symmetric types A and B, as indicated in Figure 2d,e. This is in strong contrast to the amino group functionalized [6]H on Au(111). Due to the amino functional group, which governs the molecular–substrate interaction, double chains are favored self-assembly for 5-amio [6]H to maximize the van der Waals interaction [22]. However, for primitive *rac*-[6]H, the asymmetric trimers were the favorite aggregates. These asymmetric trimers had no *C*_3_ symmetry since the distances among the three molecules were inequivalent [10]. To determine the handedness of each molecule of the trimers, we investigated the nc-AFM images at different tip-sample distances. However, we were not able to find a proper tip-sample distance to clearly determine the orientation of the outmost benzene ring for each helicene in the trimer. This was probably due to the pronounced intermolecular overlap in the trimer, which caused the helicene molecules to distort substantially. This also explained why it was a great challenge to distinguish the handedness of each molecule in chiral aggregates or assembly structures.

### 2.3. Chirality Composition of the Trimer

In order to unambiguously determine the handedness of each helicene molecule in the asymmetric trimer, we took advantage of the manipulation capability of STM to separate the trimer manually and determine the handedness of the helicene, one by one. Take one trimer A for example (the island in the overview image is NaCl, the dark spots on NaCl are CO molecules). Lateral manipulation in constant current mode was employed during the process. Upon moving the tip across the trimer (100 mV, 5 nA), it was easy to separate the trimer into one monomer and one dimer, as shown in Figure 3a–d. However, due to the strong interaction within the dimer, it was hard to divide them into separate monomers. To overcome this difficulty, the dimer was laterally moved to be attached to the edge of the NaCl island in order to pin down one helicene. Consequently, the dimer was separated into two isolated monomers by lateral manipulation, as shown in Figure 3e–h. Finally, the trimer was successfully separated into three isolated monomers, as shown in Figure 3h.

The overview STM images before and after the separation of trimer A are presented in Figure 4a and 4b, respectively. After separation, the STM and nc-AFM images were obtained of the three isolated helicene molecules one by one, which are shown in Figure 4c,d. Following the chiral determination approach, as presented in Figure 1, the descending orientations of the outmost benzene rings for all three of the isolated monomers are indicated by the corresponding arrow directions in the nc-AFM images of Figure 4d. Considering the descending orientation of the outmost benzene rings determined by the nc-AFM images, the allowed descending manner of the apparent heights of the three monomers are indicated by curved arrows in the STM images in Figure 4c. The figure shows that the monomer labeled 1 spiraled down anticlockwise, and was identified as *M*-handedness. Interestingly, the other two monomers, labeled 2 and 3, were found to have similar STM and nc-AFM images, which show the opposite descending orientation of the outmost benzene ring compared to that of monomer number 1. The allowed descending manner of the apparent heights was clockwise, so that the handedness of monomers number 2 and 3 were identified as *P*. Therefore, we clarified that the handedness of trimer A was *P*_2_*M*. Correspondingly, the handedness of trimer B should be *PM*_2_. Overall, 13 heterochiral trimers were separated and the handedness of each monomer was characterized. A total of 8 trimers were found to be type A and the other five trimers were found to be type B. In strong contrast with the previous results of *rac*-[7]H and *rac*-[5]H, which showed that the building blocks of self-assembly were heterochiral or homochiral dimers, respectively [15,17,20], this asymmetric heterochiral trimer was a novel observation in the self-assembly of helicenes, although heterochiral trimers have also been observed for [7]H/Cu(001) at very low coverage [10].

### 2.4. Theoretical Calculations

To understand why the asymmetric heterochiral trimer was the favorite chiral aggregate for *rac*-[6]H on Au(111), density functional theory (DFT) calculations, including van der Waals corrections [31,32], were performed to calculate the average binding energies for each [6]H in different aggregated configurations. Considering previous results, which showed that heterochiral or homochiral dimers were the building blocks for the self-assembly of *rac*-[7]H and *rac*-[5]H, respectively, we intentionally compared the average binding energies of homochiral and heterochiral dimers with that of heterochiral trimers. 

All the density functional theory calculations using a projector augmented wave (PAW) [33,34,35,36] method were performed using the Vienna Ab initio Simulation Package (VASP) code [37]. More accurate exchange functional optB86b-vdW [31] was employed. A periodic 10 × 12 supercell, with a three-layer slab and a vacuum thicknesses of 20 Å, was used in order to separate the slab from its periodic image in the z-direction. The bottom layer was fixed, whereas the top two layers and [6]H were allowed to relax during the optimizations. A cutoff energy of 500 eV and k-point of 1 × 1 × 1 were used [38]. The convergence criteria of the force on each relaxed atom was below 0.05 eV/Å.

The average binding energies per [6]H, representing the total molecular energy of the aggregates divided by the number of [6]H, were calculated from the following equation:$$E^{\mathrm{ave}}\left([6]H\right)=\frac{E_{\mathrm{slab}+mM-[6]H+}{{}_{n}}_{P-[6]H}-E_{\mathrm{slab}}-mE_{M-[6]H}-nE_{P-[6]H}}{m+n}$$
where m and n represent the number of *M*-[6]H or *P*-[6]H on the surface, respectively. *E*_slab + m *M*-[6]H + n *P*-[6]H_ is the total energy of the surface containing m pre-adsorbed *M*-[6]H and n pre-adsorbed *P*-[6]H, *E*_slab_ is the energy of the clean Au(111) surface. *E*_*M*-[6]H_ and *E*_*P*-[6]H_ are the energies of gaseous *M*-[6]H and *P*-[6]H, while the binding energies of a single *P*-[6]H and *M*-[6]H on Au(111) were calculated as −52.85 kcal/mol and −52.61 kcal/ mol, respectively.

The optimized binding configurations are shown in Figure 5, along with the corresponding average binding energy for each [6]H molecule. The most unstable chiral aggregate was the homochiral dimer with an average binding energy per [6]H molecule of −52.54 kcal/mol. In the case of enantiopure [6]H, the average binding energy for the homochiral tetramer was −53.41 kcal/mol, which was 0.87 kcal/mol lower than the homochiral dimer, indicating the homochiral tetramer was the favorite chiral aggregate for enantiopure [6]H. For the racemic case, the average binding energy of the heterochiral dimer was 0.95 kcal/mol lower than that of the homochiral dimer, due to the larger overlap between the two monomers in the heterochiral dimer (Figure 5a,b). Hence, it is plausible to conclude that *rac*-[6]H prefers to form the heterochiral dimer initially, instead of building the homochiral dimers. Once one additional monomer attached to the heterochiral dimer, two types of heterochiral trimers (trimer A and B) were able to form with an average binding energy of −54.29 kcal/mol, which was approximately 0.8 kcal/mol lower than that of the heterochiral dimer. Compared with previous results of *rac*-[7]H and *rac*-[5]H, the novel mode of chiral recognition in *rac*-[6]H might depend on the number of benzene rings in the helicene backbone, which allows different opening angles between the two terminal rings and results in distinct intermolecular overlap. In general, the calculated average binding energies for dimers, trimers and tetramers provided reasonable explanation of the experimental observations.

### 2.5. Racemic Hexahelicene Self-Assembly at High Coverage

The overview STM image of *rac*-[6]H on Au(111) with high coverage is displayed in Figure 6a, while the STM images at other intermediate coverage can be found in the SI. The asymmetric heterochiral trimers served as the self-assembly building blocks for different coverages, proving the stability of the aggregates [18]. Due to the reconstruction of Au(111) [39,40], the heterochiral trimers self-assembled into single and double chains in the reconstruction directions [41]. Single chains formed by one row of heterochiral trimers were found in the hcp (hexagonal-close-packed) regions, while double chains formed by one row of heterochiral trimer pairs were found in the fcc (face-centered-cubic) regions. The representative STM and nc-AFM images of the self-assembled single and double [6]H chains with high resolution are displayed in Figure 6b–e. As can be seen in the example shown in Figure 6, there was no general rule for assembling the chiral trimers and trimer pairs in the chains, where the handedness of the trimers and trimer pairs developed randomly. Nevertheless, in double chains, approximately 80% of the trimer pairs were racemic (composed of one trimer A and one trimer B), indicating that the trimers prefer to form heterochiral trimer pairs rather than homochiral trimer pairs. The detailed statistical analysis of the handedness distribution in the single and double chains can be found in the SI. These interesting results suggest that stereoselectivity also plays a role in the inter-trimer interaction, which could originate mainly from the intrinsic interaction of the helicene.

## 3. Materials and Methods

The [6]H molecules were synthesized following the reported procedure of photocyclodehydrogenation of stilbenes [23]. Enantiopure *P*-[6]H was achieved by a commercial enantioseparation process, followed by absolute configuration determination using electronic circular dichroism (CD) measurement.

Measurements were carried out with a commercial Createc LT-STM/qplus AFM (base pressure: 1 × 10^−10^ mbar). STM experiments were performed in the constant current mode at 5 K, and typical scanning parameters were 300 mV of sample bias and 10 pA of tunnelling current. The resonance frequency and oscillation amplitude of the qplus sensor were 30.7 KHz and 50 pm, respectively. The nc-AFM measurements were operated in constant height mode and with tip-sample distance changes within hundreds of pm from the set point at 300 mV, 10 pA. z < 0 means the tip was retracted from the sample from the set point. Single-crystalline Au(111) was used as the substrate and cleaned before with several cycles of Ar^+^ ions sputtering and subsequent annealing to 823 K. The [6]H samples were obtained by thermal evaporation. To prepare a CO-terminated tungsten tip, NaCl was evaporated onto the substrate. After dosing the CO, we picked the CO from the NaCl island.

## 4. Conclusions

Combining STM and nc-AFM measurements, the chiral recognition of [6]H was investigated on Au(111). A novel building block of an asymmetric heterochiral trimer was observed in the self-assembly of *rac*-[6]H. Employing the molecular manipulation capability of STM, the handedness of each helicene monomer in the asymmetric trimer was unambiguously determined one by one, demonstrating that the asymmetric trimer was of the “2+1” type of handedness. This unique trimer formation was also supported by theoretical calculations and showed a strong stability, which was in strong contrast with previous results for *rac*-[7]H and *rac*-[5]H. Our findings will promote a deeper understanding of the fundamental principles of stereochemical recognition at a molecular level.

## Figures and Tables

**Figure 1 ijms-20-02018-f001:**
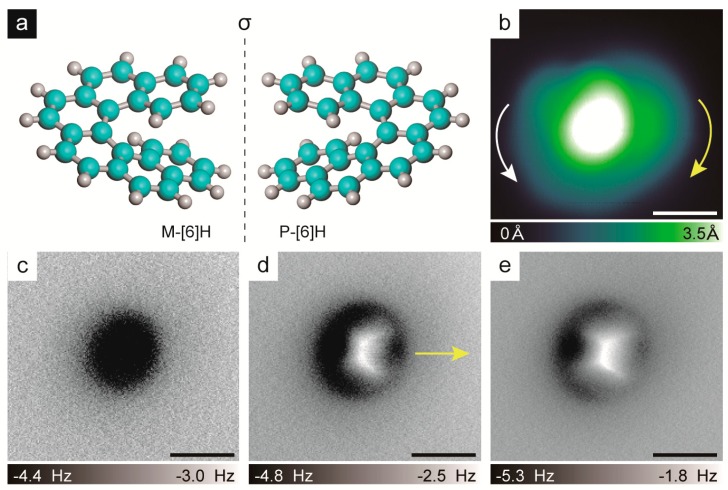
(**a**) Ball-and-stick models of left-handed hexahelicene (*M*-[6]H) and right-handed hexahelicene (*P*-[6]H), (cyan: carbon; grey: hydrogen). (**b**) Scanning tunneling microscopy (STM) image of *P*-[6]H adsorbed on Au(111) (V = 300 mV, I = 10 pA). (**c**–**e**) Noncontact atomic force microscopy (nc-AFM) images of *P*-[6]H monomer measured at different tip-sample distances of −3 Å (**c**), −2.4 Å (**d**) and −2.1 Å (**e**). Scale bars: 5 Å.

**Figure 2 ijms-20-02018-f002:**
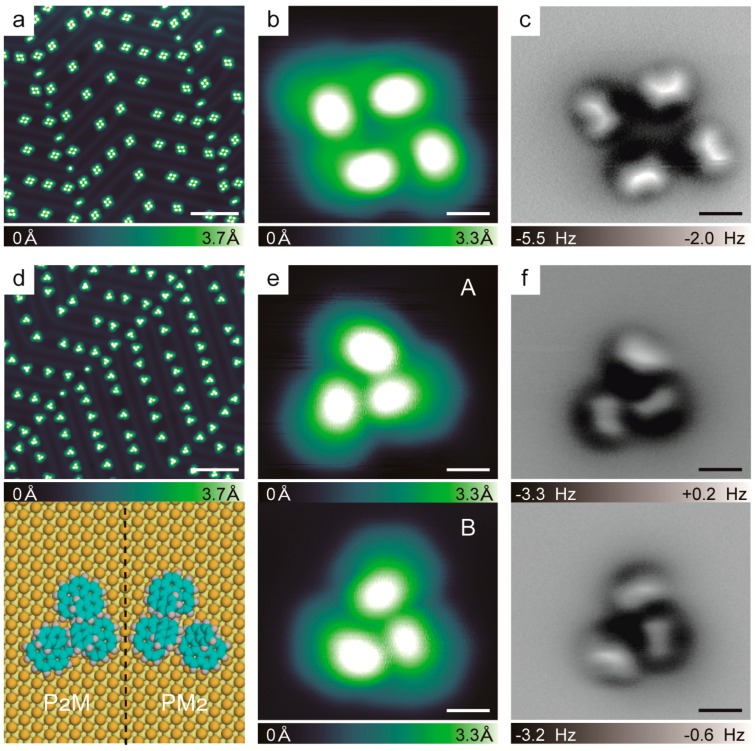
(**a**) Overview STM image of *P*-[6]H. (1000 mV, 50 pA). (**b**) Zoomed-in STM image of *P*-[6]H tetramer (300 mV, 10 pA). (**c**) nc-AFM images of *P*-[6]H tetramer measured at tip-sample distances of −3.1 Å. (**d**) Overview STM image of *rac*-[6]H (1000 mV, 10 pA) and the model of two trimers with opposite handedness, (gold, yellow: top and second/third layer of Au(111), cyan: carbon, grey: hydrogen). (**e**) Zoomed-in STM images of two types of [6]H trimers A and B (300 mV, 10 pA). (**f**) nc-AFM images of the trimer A measured at tip-sample distances of −1.6 Å and trimer B measured at tip-sample distances of −1.65 Å. Scale bars: 5 Å (**b**,**c**,**e**,**f**), 10 nm (**a**,**d**), CO tip.

**Figure 3 ijms-20-02018-f003:**
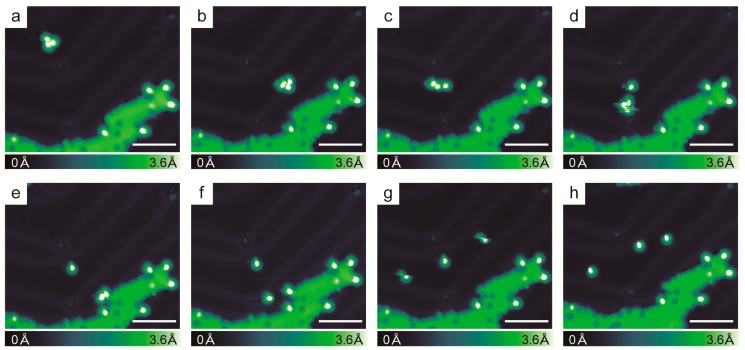
Manual separation of an asymmetric trimer into three isolated helicene monomers. (**a**–**d**) STM images of lateral manipulation process for separating the trimer into one monomer and one dimer. (**e**–**h**) STM images of lateral manipulation process for separating the dimer into two monomers. (**h**) STM image of successful manual separation of the trimer to three isolated monomers. Scanning parameters: 300 mV, 10 pA, scale bars 5 nm.

**Figure 4 ijms-20-02018-f004:**
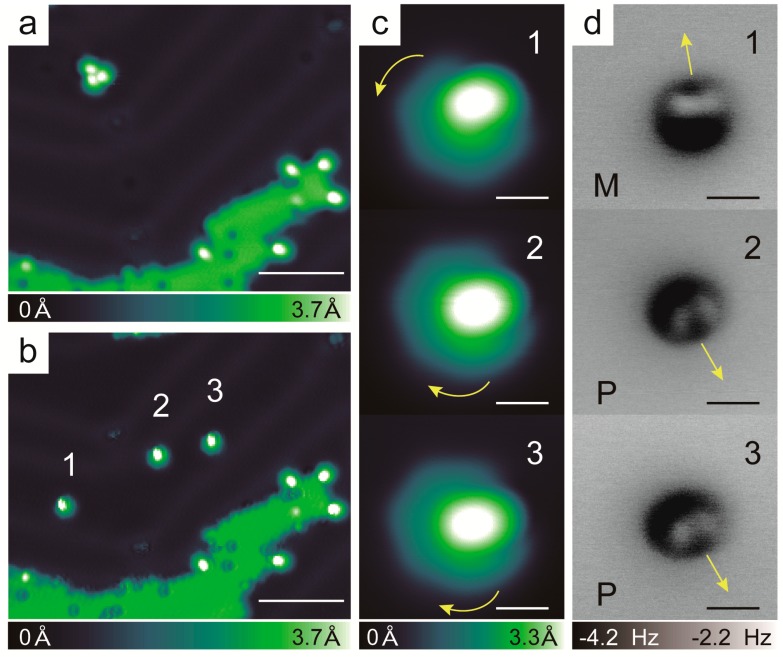
(**a**,**b**) Overview STM images before and after manual separation of the trimer (scale bars: 5 nm). The island in the overview image is a NaCl island, which was used to pick up the CO molecule for tip functionalization. (**c**,**d**) STM images, nc-AFM images of each helicene molecule manually separated from the trimer. Scale bars: 5 Å, CO tip. (300 mV, 10 pA for all STM images).

**Figure 5 ijms-20-02018-f005:**
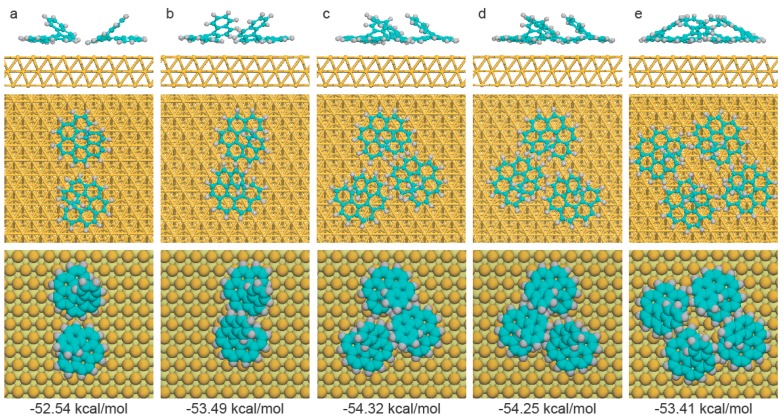
The side, top view (ball-stick model) and top view (space-filling model) of binding configurations of (**a**) homochiral dimer, (**b**) heterochiral dimer, (**c**) heterochiral trimer A, (**d**) trimer B and (**e**) homochiral tetramer, with corresponding binding energy for each [6]H labeled below.

**Figure 6 ijms-20-02018-f006:**
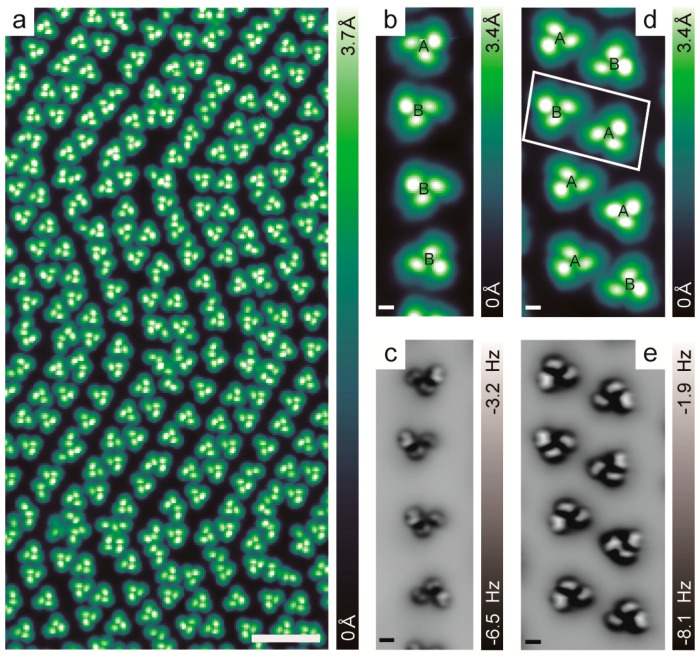
(**a**) Overview STM image of *rac*-[6]H at high coverage. (1800 mV, 10 pA). (**b**,**c**) STM and nc-AFM images of a typical single chain of *rac*-[6]H self-assembly composed of the trimers in A-B-B-B sequence. (**d**,**e**) STM and nc-AFM images of a typical double chain of *rac*-[6]H self-assembly composed of the trimer pairs in AB-BA-AA-AB sequence. One representative trimer pair BA is marked in the rectangular box. Scale bars: 5 nm (**a**), 5 Å (**b**–**e**).

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
