# Peer review of "Chiral Recognition of Hexahelicene on a Surface via the Forming of Asymmetric Heterochiral Trimers"

_ijms, 2019, doi:10.3390/ijms20082018_

Round 1
Reviewer 1 Report
In the manuscript, Zhang and coworkers observed hexahelicne ([6]H) molecules without functional moieties adsorbing on Au(111) with scanning tunneling microscopy (STM) and non-contact atomic force microscopy (AFM), and calculated the adsorption structures with van der Waals-inclusive density functional theory. They found that racemic [6]H molecules on the surface assemble dominantly into heterochiral trimers at any coverages whereas enatiopure [6]H forms heterochiral tetramers. The significant change of the building blocks depending on the enantiomer ratio is an intriguing phenomenon, and the microscopic insights would play an important role in chiral control and recognition for low-dimensional assemblies of chiral and prochiral molecules. Therefore, this paper may be published in Int. J. Mol. Sci. after considering the following points.
1) Were all the STM images shown in the manuscript obtained with CO tips? Did the tip provide better STM resolution than intact W tips? In Refs. 8 and 17, for example, the chirality of [7]H molecules can be found by STM. What determines the difficulty of chiral recognition of single molecules?
2) The authors should refer to the previous observation of heterochiral trimers of rac-[7]H on Cu(100) (Ref. 11). Unlike rac-[6]H on Au(111), the building block for rac-[7]H on Cu(100) with high coverage is not the trimers but homochiral tetramers (Ref. 18). What is the origin of the difference? The authors should also comment on the effect of the existence of functional groups (Ref. 21) to the building blocks for [6]H on Au(111).
3) It is hard to compare the molecular geometries of the trimers (Fig. 2f) with the calculated structures (bottom in Fig. 2d). The mirror plane for the calculated models (PM2 versus MP2) is aligned diagonally but that for the experimental images seems to be aligned vertically.
4) I would prefer to know the DFT-calculated binding energy of a single [6]H molecule on Au(111) so as to evaluate intermolecular interactions in the multimers.
Author Response
We thank the reviewer for considering that our results are interesting. We have modified our manuscript according to the comments of reviewers, and the comments are addressed in detail in the uploaded attachment.

Reviewer 2 Report
In this manuscript, a chiral recognition in the self-assembly of enantiopure and racemic [6]helicene on the Au(111) surface has been analyzed by using the scanning tunnelling microscopy and atomic force microscopy measurements. Authors indicated that the racemic [6]helicene molecules self-assemble in the unique the asymmetric heterochiral trimers. These observations have also been supported by theoretical calculation. The used methods and experimental data are well presented.
Some comments and suggestions:
1. Please expand the theoretical calculations section in the main manuscript because it is not clear how the theoretical calculations were performed and what results were obtained. In my opinion, Figure S6 can be placed in the manuscript for clarity.
2. Non-covalent intermolecular interactions between the [6]helicene molecules are important for formation of the heterochiral trimers and the homochiral tetramers. Your obtained intermolecular interactions and structures should be analyzed and compared with data from single crystal X-ray analysis data of [6]helicene.
The study is well-structured, and the results are of interests to a broader audience covering the readers of International Journal of Molecular Sciences. In summary, this paper is acceptable after corresponding revision.
Author Response
We thank the reviewer for appreciating our work. We have modified our manuscript according to the comments of both reviewers, and comments are addressed in detail in the uploaded attachment.
